# Homology-Directed-Repair-Based Genome Editing in HSPCs for the Treatment of Inborn Errors of Immunity and Blood Disorders

**DOI:** 10.3390/pharmaceutics15051329

**Published:** 2023-04-24

**Authors:** Daniel Allen, Nechama Kalter, Michael Rosenberg, Ayal Hendel

**Affiliations:** Institute of Nanotechnology and Advanced Materials, The Mina and Everard Goodman Faculty of Life Sciences, Bar-Ilan University, Ramat-Gan 52900, Israel; dallen413@gmail.com (D.A.); nechamaklt11@gmail.com (N.K.); michael.rosenberg@biu.ac.il (M.R.)

**Keywords:** HDR, CRISPR-Cas9, gene editing, gene therapy

## Abstract

Genome engineering via targeted nucleases, specifically CRISPR-Cas9, has revolutionized the field of gene therapy research, providing a potential treatment for diseases of the blood and immune system. While numerous genome editing techniques have been used, CRISPR-Cas9 homology-directed repair (HDR)-mediated editing represents a promising method for the site-specific insertion of large transgenes for gene knock-in or gene correction. Alternative methods, such as lentiviral/gammaretroviral gene addition, gene knock-out via non-homologous end joining (NHEJ)-mediated editing, and base or prime editing, have shown great promise for clinical applications, yet all possess significant drawbacks when applied in the treatment of patients suffering from inborn errors of immunity or blood system disorders. This review aims to highlight the transformational benefits of HDR-mediated gene therapy and possible solutions for the existing problems holding the methodology back. Together, we aim to help bring HDR-based gene therapy in CD34^+^ hematopoietic stem progenitor cells (HSPCs) from the lab bench to the bedside.

## 1. Introduction

Genome editing via targeted nucleases has fundamentally changed life-science research since it provides the ability to introduce new genes or correct mutated genes at the chosen genomic locations in a multitude of different species [1,2,3,4,5,6,7,8]. By creating a double-strand break (DSB) at a specific target site in the genome, researchers have been able to induce desired changes by harnessing the cell’s endogenous DNA repair pathways, of which there are two major mechanisms: non-homologous end joining (NHEJ) and homology-directed repair (HDR). NHEJ is an error-prone mechanism that can introduce small insertions or deletions (INDELs), leading to frameshift mutations. It has been widely used for the inactivation or knock-out of target genes [9,10]. HDR, on the other hand, requires a template, such as the sister chromatid or homologous chromosome, to repair the DSB in an error-free way, making it more appropriate for treating genetic disorders. In the context of genome editing, this has allowed researchers to utilize an exogenous donor template to actuate gene knock-in or gene correction using homologous DNA sequences via the HDR repair pathway [11].

Targeted nucleases, such as zinc-finger nucleases (ZFNs) and transcription-activator-like-effector-nucleases (TALENs) [12,13,14], have been used in the past for genome-editing purposes; however, the advent of Clustered Regularly Interspaced Short Palindromic Repeats (CRISPR)-CRISPR-associated 9 (Cas9) technology has made gene-editing applications simpler and more widely accessible due to the versatility, flexibility, and precision of the technology [4,5,15]. More specifically, in 2005, researchers were able to integrate a corrective transgene into the *IL2RG* gene via HDR with a plasmid donor template after DSB introduction by ZFNs in the K562 cell line [16]. It took another two years to adapt the capability to HSPCs [17]. The requirement for custom-made nucleases for each genomic target constituted a major hurdle while using TALENs or ZFNs, consequently impeding progress in the successful translation to gene correction in HSPCs. Despite this, specific pre-clinical and clinical trials have still been effectively conducted [18,19]. Hence, once the CRISPR-Cas9 technology was introduced to the field of genome editing, it greatly facilitated the attempts to conduct gene editing in HSPCs, especially for the purpose of ex vivo gene therapy using patients’ own hematopoietic cells. In this process, the patient’s HSPCs or T lymphocytes are isolated from the blood, edited ex vivo, and then the modified cells are transplanted back into the patient. This technique simultaneously eliminates the need to search for a human leukocyte antigen (HLA)-matched donor, which can be difficult to find, while negating the risk of graft rejection or graft versus host disease (GvHD). Thus, such a method could provide an ideal alternative to the allogeneic bone marrow transplant—the gold-standard treatment of genetic diseases of the blood and immune system—and is well suited for correcting inborn errors of immunity and blood disorders, including β-globin-related diseases such as sickle-cell disease (SCD) and β-thalassemia [20,21,22,23,24,25,26,27,28], Severe Combined Immunodeficiency (SCID) [29,30], Polyendocrinopathy Enteropathy X-linked Syndrome (IPEX) [31], Wiskott-Aldrich syndrome (WAS) [32], X-linked hyper-immunoglobulin M (hyper-IgM) (XHIM) [33], and X-linked Chronic Granulomatous Disease (X-CGD) [34]—to name a few. Previous studies and trials utilized lentiviral (LV) or gammaretroviral (γRV) vectors for ex vivo genome editing to integrate a normal copy of the gene into the genome in a semi-random fashion and transfused the cells back into the patients [35,36,37,38]. Regretfully, a number of these cases resulted in a leukemic transformation due to the activation of proto-oncogenes in the patients upon insertional mutagenesis [39,40,41,42,43]. Although significant improvements to these vectors have been made, major safety concerns still exist for LV and γRV vector-based treatments. Despite this, LV-based gene therapy for numerous forms of SCID (*RAG1*, *IL2RG*, *DCRLE1C*, and *ADA*), WAS, and other inborn errors of immunity are currently undergoing clinical trials [44,45,46,47,48,49,50,51,52,53]. To that end, a targeted CRISPR-Cas9/HDR-based methodology for site-specific transgene integration could alleviate the concerns plaguing LV and γRV vectors -incomplete phenotypic correction, dysregulated hematopoiesis, and insertional mutagenesis related to the semi-random integration and constitutive expression of the transgene [54,55].

While opportunities to apply CRISPR-Cas9-based gene therapies are vast, the current safety and efficacy barriers are inhibiting it from becoming a widely accepted method in the clinic. These include genotoxicity, toxicity associated with the exogenous DNA delivery method, insufficient HDR, low engraftment efficiency, and poor survival of transplanted cells in mice models. In this review, we aim to summarize the different methods of inducing HDR in CD34^+^ HSPCs for the purpose of correcting inborn errors of the blood and immune system. Additionally, we aim to highlight the recent and ongoing studies utilizing genome-editing HDR in HSPCs and the pitfalls currently preventing HDR-based genome editing from becoming a widely accepted clinical application. Additionally, we offer potential ways to improve the efficiency of HDR, reduce DNA-donor-delivery-associated toxicity, improve engraftment efficiencies in murine models, and increase edited cell survival. Together, these improvements will help to advance new CRISPR-Cas9-based techniques toward being applied clinically in the near future.

## 2. Gene-Editing Nucleases

TALENs, ZFNs, and CRISPR-Cas9 are all gene-editing tools that have been developed to induce targeted changes to an organism’s DNA. However, they differ in their underlying mechanisms and overall efficiency. ZFNs were one of the first gene-editing tools to be developed, and they operate via modifying zinc-finger protein domains that are fused to the DNA cleavage domain of the FokI restriction enzyme to bind to specific DNA sequences, as shown in Figure 1A. Once localized to its target locus, the ZFN nucleases produce a DSB with sticky ends [12,13,14,56,57]. Similarly, the TALEN system utilizes the FokI enzyme bound to sequence-specific TAL effector proteins [58] to induce a sticky-ended DSB at the targeted site [12,56,57] (Figure 1A). While ZFNs and TALENs can be highly specific and accurate, they are more difficult to customize, design, and implement than CRISPR-Cas9. Contrary to ZFNs and TALENs that require the production of distinct enzymes for each DNA target locus, CRISPR-Cas9 operates with a customizable guide RNA (gRNA) and a uniform nuclease (Cas9), regardless of the target. After the gRNA and Cas9 enzyme form a ribonucleoprotein (RNP) complex, the gRNA scans the DNA for a sequence that is complementary to the 20 nt spacer region with an adjacent upstream protospacer adjacent motif (PAM) sequence (5′-NGG-3′ in *S. pyogenes*) [59]. After the PAM sequence is recognized, the gRNA undergoes seed nucleation to form an RNA:DNA hybrid duplex and the now-active Cas9 nuclease induces the DSB 3 nt upstream to the PAM sequence (Figure 1B). While this review focuses specifically on HDR-based gene editing, it is important to note that the underlying mechanisms, other applications, drawbacks, and possible considerations for the different gene-editing nucleases have been extensively reviewed in Carroll et al. [12]. Additionally, while all three gene-editing tools are effective at making targeted changes to an organism’s DNA, CRISPR-Cas9 is the most widely employed in gene therapy research due to its intrinsic simplicity and versatility, as well as the massive improvements in its specificity, efficiency, and tolerability in HSPCs [60,61,62].

## 3. Mechanism of HDR

Since we will not be providing in-depth coverage of the mechanisms and different types of DNA repair involved in genome-editing applications in this review, readers may refer to Nambiar et al. [10] for a detailed analysis of this topic. Briefly, for context, once the genome-editing-induced DSB is created, a signalling cascade commences, and the cell’s DNA repair machinery is recruited to the break site. Depending on the type of HDR—canonical HDR, synthesis-dependent stranded annealing (SDSA), break-induced replication (BIR), single-stranded annealing (SSA), or single-stranded templated repair (SSTR)—the mechanism varies slightly [63,64]. In canonical HDR, the exposed DSB ends are trimmed and processed to create 3′ overhangs, while RAD51, along with other repair factors, searches for a homologous template in order to repair and refill the resected DNA. RAD51 is an ATPase that creates a nucleoprotein filament on single-stranded DNA and locates and infiltrates homologous DNA sequences for the purpose of facilitating precise DNA repair. In a native intracellular context, a homologous template comes in the form of a sister chromatid or the homologous chromosome, while in the context of gene-editing applications, it comes in the form of an exogenously introduced DNA donor template. After such a template is detected, the template undergoes strand invasion, and D-loop formation is established by annealing the overhanging tails to the homologous sequence. Once this occurs, the DNA polymerase begins to copy the sequence of the template onto the broken strand, creating a new, repaired strand of DNA. When the newly synthesized strand of DNA recognizes another homologous sequence, the elongation process is aborted, and the strand is ligated to the other exposed DSB end [65]. In gene-editing methodologies for therapeutic applications, the exogenous DNA donor template can be delivered in a number of ways including, naked single-stranded oligodeoxynucleotides (ssODNs) or longer single-stranded DNA (ssDNA), adenovirus 5/35 serotype (AdV), recombinant adeno-associated virus serotype 6 (AAV), and integration-deficient lentivirus (IDLV) (Figure 2). Since these DNA donor template platforms lack any integration machinery of their own, the DNA donors contain homology arm sequences flanking the transgene, enabling the complete integration of the entire sequence between the two homology arms into the nuclease-induced DSB. Each platform has its own benefits and drawbacks, which has led researchers to choose different methodologies for different applications, as outlined below (Table 1).

**Table 1 pharmaceutics-15-01329-t001:** Benefits and drawbacks of different DNA donor template delivery platforms.

Delivery Platform	Benefits	Drawbacks
Non-viral delivery of naked DNA	Limited DNA damage response (DDR) and/or immune response	ssODNs can only deliver relatively short sequences, thus cannot mediate gene correction via introducing a normal copy of cDNA or replacing large segments of the gene (ssDNA with Cas9 target sites could allow for introduction of larger donor sequences)
	Relatively simple to synthesize	Cannot always enable reporter gene cassettes
		Degraded quickly upon entry to cells
AdV	Superior carrying capacity (~35 kb)	Reduced HDR efficiency
	Markedly reduced extent of HDR-independent insertion into the genome due to the capped ends	
AAV	Show the highest HDR efficiencies among the currently known vectors.	Triggers a DDR proportional to the MOI used
		Limited carrying capacity (~4.8 kb)
		Can lead to unwanted off-target donor integration
IDLV	Show marked HDR efficiency	Have not been widely applied or accepted yet; and follow-up studies are being conducted to elucidate the potential benefits
	Relatively large carrying capacity (~10 kb)	
	Triggers a smaller DDR than AAV vectors	
	Less unwanted off-target integration	

## 4. Delivery of DNA Donor Template

### 4.1. Non-Viral Donor DNA Delivery

The delivery of the exogenous donor template via single-stranded oligodeoxynucleotides (ssODNs) provides a non-viral platform that enables the highly specific insertion of relatively short sequences (maximum 200 bp) [73] (Figure 2A). These viral-free vectors can be easily synthesized; however, since they are delivered as free-floating ssDNA, they degrade rapidly, reducing their efficiency [74]. ssODN integration into the genome is mediated by the RAD51-independent single-strand template repair (SSTR) pathway rather than via conventional HDR [75]. A higher HDR efficiency has been achieved by using asymmetrically designed ssODNs, namely with donor DNA that is complementary to the non-target strand [76]. ssODNs have been used to correct the *HBB* and *CYBB* genes, respectively, both in vitro (in isolated HSPCs) and in vivo (in mice) [22,26,34]. Interestingly, ssODN-mediated HDR frequencies in the *HBB* gene were found to be lower than those achieved with AAV in vitro; however, the ssODN-modified cells showed improved engraftment and survival rates when transplanted to immunodeficient mice [25,26]. A recent report demonstrated enhanced HDR in primary immune cells, including HSPCs, by designing longer ssDNA HDR templates flanked by short double-strand Cas9 target sequences that recruit CRISPR-Cas9 RNP complexes (Figure 2B). Shy et al. showed that the nuclear localization signal (NLS) on the Cas9 nuclease helped improve nuclear trafficking of the ssDNA/RNP complex, creating a favorable environment for the efficient repair of Cas9-mediated DSB by HDR using the single-strand part of the donor. In concert with HDR enhancers, they achieved the efficient correction of *IL2RA* and *CTLA4* pathogenic mutations and established GMP manufacturing practices, paving the road for potential future clinical trials [67]. While ssODNs or other non-viral platforms for DNA donor delivery are excellent ways to deliver short transcripts, these vectors are limited in their size. Therefore, they do not enable the addition of reporter gene cassettes and/or the preservation of the gene’s endogenous regulatory elements in the delivered transgene.

### 4.2. AdV

Adeno Viruses (AdVs) are vectors that can deliver a 28–36 kb non-enveloped dsDNA molecule with both 5′ ends capped with covalently attached proteins. Additionally, a high-capacity variant (Hc-AdV) with an increased carrying capacity of ~37 kb has been developed to deliver even larger transgenes [77,78,79] (Figure 2C). Aside from the ability to carry large payloads, these vectors have shown a marked benefit in the form of reduced HDR-independent transgene integration to the break sites compared to other viral vectors that carry uncapped donor DNA [80]. However, their integration efficiency is limited, yielding only 2–5% HDR following editing with ZFNs [26].

### 4.3. AAV

AAV vectors are known to be a highly efficient delivery mechanism of a donor DNA molecule to HSPCs and have been covered extensively in Dudek et al. [81]. However, these vectors have a limited carrying capacity of up to ~5 kb in size, making them applicable for the delivery of transgenes larger than ssODN, yet still unable to deliver entire cDNA for larger genes [82,83]. The ssDNA payload of the AAV donor contains inverted terminal repeats (ITRs) on both ends of the fragment that play a fundamental role in the life cycle of the virus [84]. After the ssDNA sheds its viral protein coat and enters the nucleus, it undergoes second-strand synthesis to dsDNA, with dsDNA serving as a predominant template for HDR-mediated gene targeting [69,85,86] (Figure 2D). AAV donor DNA sequences have been shown to induce high-quality HDR in CD34^+^ HSPCs with minimum homology arm lengths of ~300–400 bp, although, in certain cases, extending the homology arms to as long as 1.6–2.2 kb has shown marked improvement in HDR efficiency in HSPCs and several other primary cells [20,25,87,88]. The ease of use and early success in using AAV for transgene delivery in gene-therapy applications has boosted its popularity as a therapeutic vector. Pre-clinical studies using AAV vectors to develop potential treatments for SCD and β-thalassemia [20], SCID-X1 [30,89], WAS [32], IPEX [31], CGD [90,91,92], Hemophilia [93], XLA (X-Linked Agammaglobulinemia) [94], XMEN (X-linked Immunodeficiency with Magnesium Defect, Epstein–Barr Virus (EBV) Infection, and Neoplasia) [95], and most recently *RAG2*-SCID [29,87,96,97] have been conducted in CD34^+^ HSPCs, showing highly efficient frequencies of HDR. Interestingly, most AAV donor structures have the homology arms flanking the immediate genomic sequences 5′ and 3′ to the Cas9-induced DSB, leading to the insertion of the transgene into the break site. However, recent works have shown that distancing the 3’ homology arm from the DSB site allows for the replacement of up to several kilobases of genomic DNA with the transgene [20,87] (Figure 3). Allen et al. utilized this replacement strategy to replace the entire endogenous *RAG2* coding sequence, allowing for transgenes to maintain critical endogenous regulatory elements while preserving the integrity and 3D architecture of the genomic locus as much as possible by limiting the introduction of kilobases of new DNA [87]. This replacement strategy is particularly useful for correcting tightly controlled genes, such as *RAG2*, that are regulated by spatiotemporal elements in the surrounding locus [98]. Additionally, this replacement strategy was used by Cromer et al. to replace the *α-globin* (*HBA*) gene with the *β-globin* (*HBB*) gene to restore the balance in β-thalassemia-patient-derived HSPCs [20]. While AAVs are non-pathogenic and elicit a low immune response, they are still capable of triggering a toxic DDR proportional to the amount of virus used or the multiplicity of infection (MOI) [99]. Thus, finding the delicate balance between inducing high frequencies of HDR while limiting the MOI used is critical. Unfortunately, a recent Phase 1/2 clinical study by Graphite Bio for the treatment of SCD was paused due to the development of pancytopenia in the first patient treated with their AAV-based therapy [100], highlighting the importance of finding this balance.

### 4.4. IDLV

To circumvent the limited carrying capacity of AAV vectors, some researchers have utilized co-delivery dual-vector approaches, namely splitting the transgene between two distinct AAV vectors that are transduced simultaneously [101,102]. IDLV vectors could provide a more straightforward alternative to AAV when larger payloads are desired. IDLVs are lentiviral vectors packaged with a catalytically inactive integrase, allowing for effective transduction while limiting genomic integration since the genomic payload remains episomal. These vectors carry the donor sequence in the form of RNA, which is converted into dsDNA with exposed ends via reverse transcription upon entry into the target cell. IDLVs have a significantly larger carrying capacity than AAVs (~10 kb) and alleviate the safety concerns regarding the semi-random integration of standard LVs since they are expressed transiently and lack the integration mechanisms of traditional LV vectors [72,103,104,105] (Figure 2E). The IDLV system could provide a valuable tool for the replacement of larger genes, potentially increasing the chance of a functional gene correction with minimal interference to the endogenous locus structure and spatiotemporal regulatory elements. To date, studies in HSPCs using IDLV-based HDR to correct SCID-X1- and SCD-causing mutations have shown encouraging results when paired with ZFN or CRISPR-Cas9 [17,19,26,103,106].

## 5. Existing Barriers to Efficient HDR-Based Therapies

### 5.1. Specificity

Despite the ability of genome-editing nucleases to induce site-specific targeted editing, undesired on- and off-target editing (and the effects associated with it) remain a significant issue [107,108]. In the case of CRISPR-Cas9, off-target editing occurs as a result of mismatched bases between the gRNA and DNA sequence being tolerated and still inducing a Cas9 DSB. Since the CRISPR-Cas9 system originated in bacteria, this flexibility is believed to be a beneficial evolutionary adaptation for immunity against previously encountered mutated bacteriophages [61]. Accompanying the induction of DSBs at off-target sites is the potential risk of inducing structural variations (translocations, inversions, insertions, and deletions) within edited sites, which can be particularly hazardous [109,110,111,112,113,114,115,116,117]. Numerous platforms and approaches have been developed for the purpose of detecting these off-target sites and structural variations [110,118,119,120]. Additionally, the unwanted integration of the exogenous HDR template into off-target edited sites has been reported [121]. While many studies have been conducted to both limit and quantify the off-target activity of the gene-editing nucleases [61], a recent study by Ferrari et al. showed that the use of IDLV showed less unwanted donor integration to off-target sites when compared to AAV vectors [121]. Additionally, genotoxicity stemming from the presence of unintended and unwanted genomic modifications at the on-target locus has been reported as an issue that, unlike the risk associated with off-target editing, cannot be alleviated via more specific DSB approaches. These genomic aberrations can vary from a few bp to the kilobase- or even megabase-scale [122].

### 5.2. Insufficient HDR/Bias towards NHEJ

Since HDR occurs only during the S/G_2_ phase of the cell cycle, in the slow-dividing quiescent HSPCs, there is often a bias towards the NHEJ repair mechanism [10,123]. While highly efficient HDR is a pre-requisite for any succsessful clinical application, depending on the disorder being treated, the threshold for the number of cells that need to have undergone successful HDR in the ex vivo editing process varies substantially [124]. For example, with SCID, a relatively small number of corrected cells (~1000 HSCs), if engrafted properly, could reconstitute the entire immune system [125,126,127]. Additionally, based on the reported cases of “naturally occurring gene therapy”, in which a revertant mutation in a single cell enabled T lymphocyte survival and the development of diverse TCR repertoires in a SCID patient [125,128,129,130], it is evident that a minuscule number of “corrected” cells can reconstitute the immune system and abrogate the SCID phenotype. Moreover, Dvorak et al. demonstrated that donor chimerism as low as 3% was sufficient to reconstitute T- and B-cell populations in SCID patients, while for diseases such as thalassemia or SCD, >20% is required [131]. For this reason, cases of suboptimal HDR editing efficiency can lead to the downfall of certain therapeutic applications. Thus, the development of methods to boost HDR frequency is critical. To address these challenges, researchers have experimented with Cas9/HDR template conjugates [61] and have also investigated the potential use of molecules, including i53 and/or DNA-PK inhibitors, to inhibit the NHEJ repair pathway and skew the repair mechanisms towards the HDR repair pathway. Through the use of these molecules, one can transiently inhibit the NHEJ repair pathway and control the cycling and quiescence of the edited cells. This has been shown to significantly increase HDR integration frequencies [67,75,132,133,134]. Additionally, recent studies have shown success in tipping the scale towards HDR by inhibiting DNA polymerase θ (Polθ), a key effector in the DSB repair pathway, by microhomology-mediated end-joining (MMEJ) [10,135,136]. While the stable knock-out or knock-down of DDR and repair proteins can produce irreversible and dangerous effects, utilizing these small molecules can have a transient knock-down effect in the short time window wherein gene editing and HDR occur. In addition to these techniques being efficient methods to enhance HDR frequencies, they can also contribute to reducing virus-associated toxicity by reducing viral load while maintaining highly efficient HDR. Additionally, Shin et al. demonstrated that HDR could be increased through controlled cycling and quiescence of the edited cells [106,137]. Lastly, when HDR efficiencies are low, extending the homology arms of the donor DNA to as long as 1.6–2.2 kb has shown a marked improvement in HDR frequencies via the use of AAV in HSPCs and several other primary cells [20,25,87,88].

### 5.3. Toxicity

In addition to genotoxicity associated with undesirable DNA repair outcomes resulting from Cas9-induced DSBs, HSPCs are very sensitive to DNA damage both at the on and off-target sites [10]. Additionally, HSPCs have innate immune cues and a plethora of antiviral factors and nucleic acid sensors that are activated in the process of exogenous gene delivery [138,139]. Since all current HDR-based gene-editing technologies expose HSPCs to some form of exogenous nucleic acid, the resulting immune response is of significant concern to researchers and clinicians alike. Additionally, AAV and IDLV vectors are both recognized by DDR proteins in HSPCs, triggering a DDR [99,124,140,141]. More specifically, the transduction of these viral vectors triggers a p53 response, leading to reduced HSPC proliferation, apoptosis, and poor transplantation to immunodeficient mice [106,141]. While the exact DDR mechanism has yet to be elucidated, Allen et al. showed that the level of the DDR was directly proportional to the MOI used [99]. Moreover, Iancu et al. showed that using high viral load led to prolonged viral presence in the cell population, decreased cell yield, and positive selection of unedited cells, leading to a consistent reduction of HDR frequencies over the course of T-cell differentiation in vitro. To alleviate this effect, Iancu and colleagues demonstrated that reducing the amount of viral vector enabled better HSPCs survival and T-cell differentiation of corrected SCID-patient cells in vitro [97]. Interestingly, transient p53 inhibition using a dominant negative p53 truncated form (GSE56) [142] has been shown to significantly preserve the transplanted HSPCs’ multilineage potential [106,141]. Unmitigated AAV-induced toxicity can explain, at least to some extent, the pancytopenia phenotype that occurred in the first AAV-mediated HDR-based clinical trial experiment for the correction of the *HBB* gene, which has since been halted [100]. Thus, fine-tuning viral-delivery systems to reduce toxicity by reducing the viral load uses and increasing HDR frequencies is critical. In contrast to AAV and IDLV vectors, ssODNs do not elicit p53 activation and are relatively well-tolerated by HSPCs [25,26]. It should be noted that a recent study by Ferrari et al. showed that, when comparing AAV and IDLV vectors directly, the latter induced both a less persistent DDR as well as lower unwanted donor integration, leading to improved clonogenic capacity and editing efficiency in long-term HSPCs [121]. While this would be a great step towards clinical translation in the form of HSPC gene therapy, further studies are required to confirm these results.

### 5.4. Poor Engraftment of Edited Cells

While gene-editing technologies and transgene delivery mechanisms have made a lot of progress in the past few years, poor engraftment frequencies of the LT-HSC-edited population in the bone marrow of immunodeficient mice [139] has been a recurring problem in proof-of-concept pre-clinical studies when the exogenous template being used for integration is delivered in a virus-dependent manner in CD34^+^ HSPCs [25,26,50,143]. In addition to the transgene-induced stress, HSPCs have been shown to lose their stemness and engraftment potential when exposed to extensive culturing protocols. This process is a product of poor cell fitness, impairing their ability to reach the bone marrow niche with subsequently reduced engraftment efficiencies [139,144,145,146,147]. Specifically, with the use of the AAV vector, high MOI is still required to reach clinically relevant levels of HDR. Since the toxic cellular responses mentioned above can account for the poor engraftment capabilities [139], the development of strategies to limit the viral dose while maintaining the HDR frequencies is crucial [75,132,133,137]. Additionally, Iancu et al. highlighted the possibility of enriching HDR-positive cells in order to limit the competition in vitro with HDR-negative cells that might prevail [97]. In that study, SCID patient cells were treated with CRISPR-Cas9 and AAV vectors carrying a corrective transgene, and both sorted HDR^+^ cells and unsorted cells were differentiated in vitro. While the unsorted corrected cells failed to efficiently differentiate and proliferate, the sorted population thrived and developed into CD3^+^ T cells with diverse T-cell receptor (TCR) repertoires [97]. While this was only performed in vitro, transplanting a homogenous enriched HDR-positive population could improve the engraftment frequencies in vivo, and follow-up studies in mice are required to elucidate this strategy further. Additionally, Poletto et al. showed that improved engraftment of AAV-edited CD34^+^ HSPCs in immunodeficient mice was possible via Busulfan-based myeloablation [148]. A study by Pavel-Dinu et al. also showed high human chimerism following the in vivo transplantation of cells edited with a WT cDNA cassette targeting the *IL2RG* locus [30]. This effect could be attributed to the substantial quantity of cells employed in this study, more than 2 × 10^6^ cells per mouse, which may have been enough to offset the toxicity-related cell death and still result in the functional engraftment of the edited population. Albeit, while the engraftment was an overall success, the frequency of human cells was substantially reduced between 1° and 2° mice, which is indicative of reduced cell survival of the edited cells and/or competition with the unedited cells for the bone marrow niche. This issue can possibly be abrogated by (1) limiting the viral vector, and thus the viral-vector-induced toxicity; (2) increasing the input HDR, increasing the chances for the edited cells to compete for the bone-marrow niche; (3) using FACS enrichment of the edited cells before transplantation, limiting niche competition even further; (4) transplanting a greater number of cells, allowing for greater chances of overall engraftment; and/or (5) using ssODNs, which have shown better outcomes in vivo compared to the viral delivery of the transgene.

## 6. Conclusions

Ex vivo genome editing holds tremendous promise as a therapeutic strategy for monogenic disorders of the blood and immune system. HDR-based platforms offer a versatile approach to functional gene correction, allowing for the addition of curative transgenes to the endogenous locus. Additionally, strategies have been developed to completely remove mutated transcripts and replace them with corrective transgenes, which would allow for the conservation of important regulatory and spatiotemporal elements. Many of these targeted approaches are highly relevant for treating diseases in which possible disease-causing mutations are dispersed over the entirety of the gene, and a universal solution to treat any possible mutations in the given gene cannot be provided using NHEJ-based methodologies (Table 2). Such a HDR-mediated option could permit correcting all possible mutations in a given gene with a single corrective sequence, eliminating the need for patient-by-patient customization. While HDR-based gene therapies have yet to successfully reach the clinic and early results from clinical trials have had some challenges, the potential of these strategies is still tremendous. This review outlines several donor DNA delivery platforms available for HDR-mediated gene correction, including both viral-free and viral-dependent delivery systems, as well as the current drawbacks preventing each of them from being widely accepted in clinical applications. Additionally, we aim to highlight potential ways to alleviate these safety and efficacy concerns to help push these technologies toward a reliable and safe treatment. While research is currently being conducted to solve these major issues and the field of ex vivo genome editing continues to progress, the development of more precise and efficient editing tools will be key in translating HDR-based genome editing for the treatment of inborn errors of immunity, diseases of the blood system, and other genetic disorders.

**Table 2 pharmaceutics-15-01329-t002:** HDR gene-editing studies for the correction of disorders of the blood and immune system, performed in HSPCs, using ZFNs, TALENs, and Cas9. In vitro refers to cell culture studies and in vivo refers to the transplantation of edited HSPCs to animal models.

	Targeted Gene	Nuclease	Donor Platform	Study	Reference
SCID	*WAS*	Cas9	AAV	in vivo	[32]
	*RAG2*	Cas9	AAV	in vitro	[87,96,97]
		Cas9	AAV	in vivo	[29]
	*IL2RG*	ZFN	IDLV	in vitro	[17]
		ZFN	IDLV	in vivo	[103,134]
		ZFN	AAV	in vivo	[134,141]
		Cas9	IDLV	in vivo	[134]
		Cas9	AAV	in vitro	[89]
		Cas9	AAV	in vivo	[30,89,134,141]
CGD	*AAVS1*	Cas9	AAV	in vivo	[90]
	*CYBB*	Cas9	ssODN	in vivo	[34]
		Cas9	AAV	in vivo	[91]
	*NCF1*	ZFN	AAV	in vivo	[92]
IPEX	*FOXP3*	Cas9	AAV	in vivo	[31]
SCD/ β-thalassemia	*HBB*	ZFN	IDLV	in vivo	[19,26]
		ZFN	ssODN	in vitro	[23]
		ZFN	ssODN	in vivo	[19,26]
		ZFN	AdV	in vivo	[26]
		ZFN	AAV	in vivo	[26]
		TALEN	ssODN	in vitro	[23]
		Cas9	IDLV	in vitro	[24]
		Cas9	IDLV	in vivo	[26]
		Cas9	ssODN	in vitro	[23]
		Cas9	ssODN	in vivo	[22,25,26,28]
		Cas9	AAV	in vivo	[20,21,25,26,27,88]
		Cas9	AdV	in vivo	[26,79]
Hemophilia	*HBA*	Cas9	AAV	in vivo	[93]
XHIM	*CD40L*	TALEN	IDLV	in vivo	[33]
		TALEN	AAV	in vivo	[33]
		Cas9	IDLV	in vivo	[33]
		Cas9	AAV	in vivo	[33]
XLA	*BTK*	Cas9	AAV	in vivo	[94]
XMEN	*MAGT1*	Cas9	AAV	in vivo	[95]

## Figures and Tables

**Figure 1 pharmaceutics-15-01329-f001:**
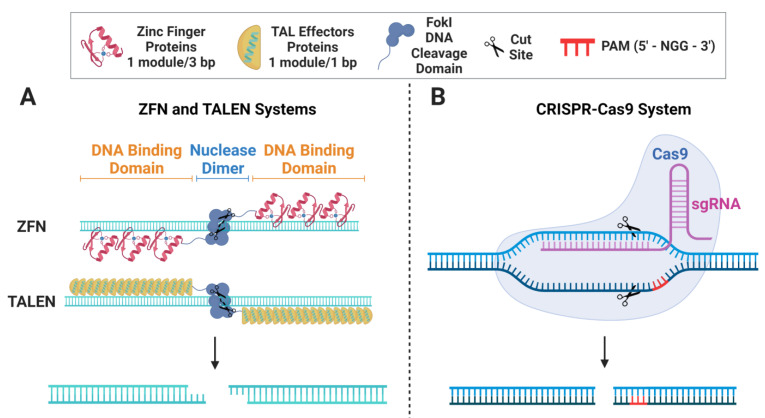
**Genome-editing Nucleases.** (**A**) Schematic depicting ZFN and TALEN genome-editing systems and the resulting DSB. (**B**) Schematic depicting the CRISPR-Cas9 genome-editing systems and the resulting DSB.

**Figure 2 pharmaceutics-15-01329-f002:**
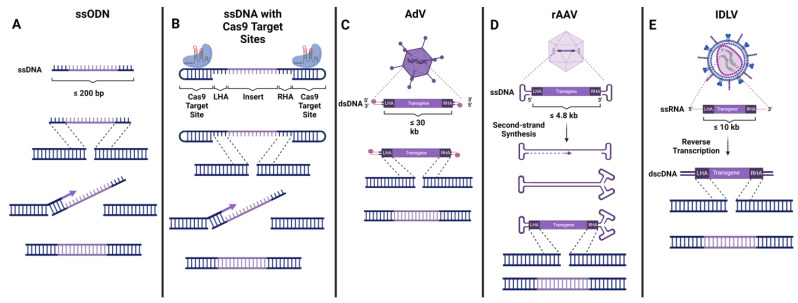
**Delivery Mechanisms for HDR.** (**A**) Single-stranded oligodeoxynucleotides (ssODNs). Together with ssODNS, DSBs are processed by the RAD51-independent SSTR pathway, in which the ssDNA filament invades the homologous region to form a D-loop structure, enabling DNA synthesis by a polymerase and eventual resection of the DSB [66]. (**B**) ssDNA with dsDNA Cas9 target sequences on the ends. These donors have been shown to induce efficient HDR at donor lengths of up to ~3.5 kb. The NLS on the Cas9 allows for more efficient trafficking of the donor into the nucleus, enabling enhanced HDR efficiencies [67]. (**C**) Adenovirus 5/35 (AdV). AdVs are capable of carrying very large dsDNA payloads of ~30–35 kb. The 5′ termini of AdV genome bind covalently to virus-coded terminal proteins, enabling better stability and markedly reduced HDR-independent incorporation into the genome [68]. (**D**) Recombinant adeno-associated virus serotype 6 (AAV). AAV vectors carry an ssDNA payload with inverted terminal repeats that create dsDNA loops at both the 5′ and 3′ ends of the genome. After entering the cell, the genome is believed to undergo second-strand synthesis, providing a dsDNA intermediate to facilitate HDR and integration into the genome [69,70,71]. (**E**) Integration-deficient Lentivirus (IDLV). IDLVs carry single-stranded viral RNA along with other enzymes including reverse transcriptase. When the ssRNA payload is released into the cytoplasm it undergoes reverse transcription, producing a dsDNA product that can then be imported into the nucleus to serve as an HDR template [72].

**Figure 3 pharmaceutics-15-01329-f003:**
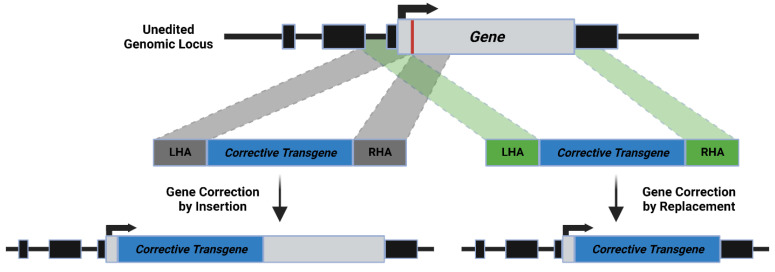
**HDR formulations: Insertion vs. Replacement.** Both techniques utilize homology arms; however, whereas the insertion method (gray dotted lines) has homology arms centered at the Cas9 cut site (depicted as the red line downstream to the gene’s ATG), the replacement method (green dotted lines) has a left homology arm flanking the genomic DNA upstream to the cut site while the right homology arm is distanced to be downstream to the region being replaced, in the depicted case, downstream to the stop the codon of the gene’s open reading frame (ORF).

## Data Availability

No new data were created or analyzed in this study.

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
