# Peer review of "Homology-Directed-Repair-Based Genome Editing in HSPCs for the Treatment of Inborn Errors of Immunity and Blood Disorders"

_pharmaceutics, 2023, doi:10.3390/pharmaceutics15051329_

Round 1

Reviewer 1 Report

Allen et al have provided an extensive review of the current status of the field. The review reads well and is timely. I have below suggested some segments that can be added for completeness, and some other sections that I would suggest omitting since these do not add to the storyline andonly deviate from the message stipulated in the title. In my opinion the text could be improved by an additional proof reading, leaving out subjective wording and correcting some wording (see comments below for examples). 

General comments:

The authors discuss the delivery platforms for the the DNA template to allow HDR. It would be helpful to include a paragraph or subsection commenting on the delivery platforms available for Cas9 or CRISPR machinery. 

Table 1: Please include the delivery method for the Cas9 components. RNPs, transfection, electroporation, VLPs,…

Table 2: please remove subjective wording (very large, very high, low). It is not clear what this actually means. For the AAV approach it is also reported that the ssDNA AAV is more efficient than de scAAV (https://www.ncbi.nlm.nih.gov/pmc/articles/PMC6927556/)

Line8: CRISPRCas did not revolutionize the world of gene therapy. I reckon for a review on this topic, it is important to state where we are, and at the moment this statement is not correct in my opinion. I would like to suggest to be more moderate, and either state ‘the field of GT research’, or ‘is expected to revolutionize/disrupt’.

Line9: Again, for me it is important to not polarize the field. GT has the potential to cure disease, but frequently it does not, it halts a disease or slows it down, which is in essence also remarkable. I would suggest describing more moderately.

Line15: sentence does not read well (when considered for treating?)

Line17: holding in order? Please rephrase.

Line 44: This is not correct, please verify. The gene was not inserted, ZFN cut at the site and through recombination with the sister chromatid. (also in Fig1)

Line67 and 77: retroviral vector do not integrate in a semi-random fashion. gRV integrate preferably in near enhancers and promoter regions, whereas LV integrate in the body of active genes (see . Maybe it is better to refer to https://pubmed.ncbi.nlm.nih.gov/31951833/, https://pubmed.ncbi.nlm.nih.gov/26293289/.

Line141: FokI nuclease is not correct. Better refer to this as a FokI domain (2 ZF-FokI monomers must bind to their respective ZF DNA-binding sites on opposite strands in an inverted orientation, usually separated by 5–7 nt, in order to form a catalytically active dimer).

Line159: widely employed in which field? GT research?

Line 223: ‘transduction’ is not correct here (the delivery is viral vector-free). 

Lines 226-227: it is not necessary to get the second strand synthesized to enable the HDR repair. 

Line259: all AAV can be used, not only AAV6 pseudotyped vectors. In addition, in this section the authors elaborate on topics beyond the scope of this review. I would leave out all info on rAAV gene therapy. Also, the authors discuss AAV virus and rAAV vectors which complicates the message. For example, “rAAV6s are not pathogenic” (line296), this is not correct. rAAV refer to viral vectors, and these are by definition not pathogenic, it is the AAV virus that is not pathogenic. The part on Glybera is not relevant here.

Line295: “bthal derived HSPC” this wording is not correct. The HSPC are not derived from bthal.

Line300: I would leave out the info on the GT trial mentioned here (not relevant, since the GT trail is about in vivo rAAV therapy)  

Fig3: this figure is not specific to rAAV6, so I would suggest adjusting the title. Also in the legend ‘the gene’s stop codon’. This is not correct, this should read ‘the stop codon of the gene’s ORF’.

Line311: explain what IDLV are (mutated in IN)

Lines 331-340: it may be worth mentioning the type of platforms and approaches that are available to detect those structural variations and off-targets (more relevant to me than the Glybera story) 

Line 354: typo ‘sucsesful’ > ‘successful’

Line 391 and following: it is difficult to follow why immunity is discussed (referring to papers on complement ref140). Here more detail is required in my opinion. HSPC based therapy in essence is performed ex vivo, and thus only innate immunity in the cell should be considered. It may be interesting to have an additional paragraph looking at the future, where this technology would be applied in vivo, however, this would be not the most efficient approach for HSPC and blood disorders I reckon. 

Line 393 and on: As indicated higher up, it would be interesting to also include a paragraph on the delivery of the ZNF, TALENs, CRISPR machinery. For example, for the topics discussed here, may the use of viral-like particles (VLPs - https://www.nature.com/articles/s41467-018-07845-z, https://pubmed.ncbi.nlm.nih.gov/35021064/) vectors to deliver Cas9 reduce the toxicity of the approaches (section 5.3)? Might this affect the engraftment ability of the HSPCs as mentioned in 5.4?

Line411: ‘virus used’, this should be viral vector 

line463: viral load, virus-induced (should be viral vector)

Line 471: typo ‘ex vivo’ > ‘Ex vivo

Line 479: “universal solution cannot be provided using NHEJ-based methodologies”. What is meant with ‘universal’, do you mean generic? Please check Nami et al., 2018: https://doi.org/10.1016/j.tibtech.2018.03.004

Author Response

Reviewer 1:

General comments:

The authors discuss the delivery platforms for the the DNA template to allow HDR. It would be helpful to include a paragraph or subsection commenting on the delivery platforms available for Cas9 or CRISPR machinery. 

Table 1: Please include the delivery method for the Cas9 components. RNPs, transfection, electroporation, VLPs,…

We appreciate the reviewer’s suggestion. Unfortunately, since our manuscript is devoted to homology-directed repair (HDR) in HSPCs and, given the stringent space limitation, we decided to leave the delivery platforms for the CRIPSR machinery per se out of the scope of this review and focused solely on delivery platforms for HDR donors.

Table 2: please remove subjective wording (very large, very high, low). It is not clear what this actually means. For the AAV approach it is also reported that the ssDNA AAV is more efficient than de scAAV (https://www.ncbi.nlm.nih.gov/pmc/articles/PMC6927556/)

Following the reviewer’s suggestion, we removed the subjective wording and replaced it with more accurate definitions. Regarding scAAV, a previous study showed a loss of detectable editing with scAAV (PMID: 16822856), so we left it out of the scope of this manuscript. 

Line8: CRISPRCas did not revolutionize the world of gene therapy. I reckon for a review on this topic, it is important to state where we are, and at the moment this statement is not correct in my opinion. I would like to suggest to be more moderate, and either state ‘the field of GT research’, or ‘is expected to revolutionize/disrupt’.

We agree with the reviewer’s suggestion and have changed the wording to, "the field of gene therapy research".

Line9: Again, for me it is important to not polarize the field. GT has the potential to cure disease, but frequently it does not, it halts a disease or slows it down, which is in essence also remarkable. I would suggest describing more moderately.

We agree with the reviewer’s suggestion and have changed the wording here as well

Line15: sentence does not read well (when considered for treating?)

We have changed the wording to,"yet all possess significant drawbacks when applied in the treatment of patients suffering from inborn errors of immunity or the blood system"

Line17: holding in order? Please rephrase.

We have changed the wording to,"Together, we aim to help bring  march HDR-based gene therapy in CD34+ hematopoietic stem progenitor cells (HSPCs) from the lab bench to the bedside."

Line 44: This is not correct, please verify. The gene was not inserted, ZFN cut at the site and through recombination with the sister chromatid. (also in Fig1)

We agree with the reviewer that the transgene was not inserted but rather, to our knowledge, was copied from the plasmid template during the HDR process, in the original paper (PMID: 15806097). We have clarified it in the text. Figure 1 was used merely to describe the HDR-mediated mechanism of DSB repair which, in native conditions, utilizes a sister chromatid as a recombination template, while in the experimental setting can utilize exogenous donor template, such as plasmid DNA.

Line67 and 77: retroviral vector do not integrate in a semi-random fashion. gRV integrate preferably in near enhancers and promoter regions, whereas LV integrate in the body of active genes (see . Maybe it is better to refer to https://pubmed.ncbi.nlm.nih.gov/31951833/, https://pubmed.ncbi.nlm.nih.gov/26293289/.

While we concur with the reviewer’s opinion that the semi-random mode of lentiviral integration is a simplification, we could not provide a more detailed explanation of the integration tendencies due to space restrictions and the scope of this manuscript.

Line141: FokI nuclease is not correct. Better refer to this as a FokI domain (2 ZF-FokI monomers must bind to their respective ZF DNA-binding sites on opposite strands in an inverted orientation, usually separated by 5–7 nt, in order to form a catalytically active dimer).

We are thankful to the reviewer for pointing it out and we revised the text as follows "ZFNs were one of the first gene-editing tools to be developed, and they operate via modifying zinc-finger protein domains, fused to the DNA cleavage domain of the FokI restriction enzyme, to bind to specific DNA sequences, as shown in Figure 1A.  Once localized to its target locus, ZFN nucleases produce a DSB with sticky ends". Additionally, we changed Figure 1 to refer to the FokI region as the "FokI DNA cleavage domain".

Line159: widely employed in which field? GT research?

We agree with the reviewer’s suggestion and have changed the wording to, "in gene therapy research"

Line 223: ‘transduction’ is not correct here (the delivery is viral vector-free). 

We thank the reviewer for this comment and have removed the word from the text

Lines 226-227: it is not necessary to get the second strand synthesized to enable the HDR repair. 

Indeed, the recombination mechanism might theoretically involve either a single-stranded or double-stranded template. However, according to Kan et al, (PMID: 24699519), the HDR mechanism that utilizes a double-stranded template was found to be a predominant one in AAV-mediated gene targeting. We clarified it in the text.

Line259: all AAV can be used, not only AAV6 pseudotyped vectors. In addition, in this section the authors elaborate on topics beyond the scope of this review. I would leave out all info on rAAV gene therapy. Also, the authors discuss AAV virus and rAAV vectors which complicates the message. For example, “rAAV6s are not pathogenic” (line296), this is not correct. rAAV refer to viral vectors, and these are by definition not pathogenic, it is the AAV virus that is not pathogenic. The part on Glybera is not relevant here.

We thank the reviewer for this suggestion and have changed the references to AAVs to be more clear. Additionally, we agree with the suggestion that the portion about Glybera and Luxturna are not relevant here and have adapted the text accordingly.

Line295: “bthal derived HSPC” this wording is not correct. The HSPC are not derived from bthal.

We agree with the reviewer and modified the text accordingly, although it was the original wording used by Cromer et al (PMID: 33737751).

Line300: I would leave out the info on the GT trial mentioned here (not relevant, since the GT trail is about in vivo rAAV therapy)  

We appreciate the reviewer’s concern, however, would like to clarify that indeed the referenced GT trial (ClinicalTrials.gov Identifier: NCT04819841) is in fact referring to ex vivo editing of a patients CD34+ HSPCs for the purpose of autologous transplantation. That being said, we wish to include this incident as a cautionary example of the possible risks associated with the current method of gene editing with current methods.

Fig3: this figure is not specific to rAAV6, so I would suggest adjusting the title. Also in the legend ‘the gene’s stop codon’. This is not correct, this should read ‘the stop codon of the gene’s ORF’.

We agree with the reviewer’s suggestion and have changed the text accordingly

Line311: explain what IDLV are (mutated in IN)

We appreciate the reviewer’s suggestion and have added the following sentence to clariy: “IDLVs are lentiviral vectors packaged with a catalytically inactive integrase, thus allowing for effective transduction while limiting genomic integration since the genomic payload remains episomal”.

Lines 331-340: it may be worth mentioning the type of platforms and approaches that are available to detect those structural variations and off-targets (more relevant to me than the Glybera story) 

We agree with the reviewer’s suggestion and have added the following sentence with relevant citations: “Numerous platforms and approaches have been devloped for the purpose of detecting these off-target sites and structural variations.”

Line 354: typo ‘sucsesful’ > ‘successful’

We thank the reviewer for this catch and have corrected the mistake

Line 391 and following: it is difficult to follow why immunity is discussed (referring to papers on complement ref140). Here more detail is required in my opinion. HSPC based therapy in essence is performed ex vivo, and thus only innate immunity in the cell should be considered. It may be interesting to have an additional paragraph looking at the future, where this technology would be applied in vivo, however, this would be not the most efficient approach for HSPC and blood disorders I reckon. 

We agree with the reviewer and removed the aforementioned sentence since it has indeed been irrelevant to the ex-vivo treatments.

Line 393 and on: As indicated higher up, it would be interesting to also include a paragraph on the delivery of the ZNF, TALENs, CRISPR machinery. For example, for the topics discussed here, may the use of viral-like particles (VLPs - https://www.nature.com/articles/s41467-018-07845-z, https://pubmed.ncbi.nlm.nih.gov/35021064/) vectors to deliver Cas9 reduce the toxicity of the approaches (section 5.3)? Might this affect the engraftment ability of the HSPCs as mentioned in 5.4?

We appreciate the reviewer’s suggestion. Unfortunately, since our manuscript is devoted to homology-directed repair (HDR) in HSPCs and, given the stringent space limitation, we decided to leave the delivery platforms for the CRIPSR machinery per se out of the scope of this review and focused solely on delivery platforms for HDR donors.

Line411: ‘virus used’, this should be viral vector 

We agree with the reviewer’s suggestion and have changed the text accordingly

line463: viral load, virus-induced (should be viral vector)

We agree with the reviewer’s suggestion and have changed the text accordingly

Line 471: typo ‘ex vivo’ > ‘Ex vivo

We thank the reviewer for this catch and have corrected the mistake

Line 479: “universal solution cannot be provided using NHEJ-based methodologies”. What is meant with ‘universal’, do you mean generic? Please check Nami et al., 2018: https://doi.org/10.1016/j.tibtech.2018.03.004

We appreciate the reviewer's comment and have adapted the wording to clarify: "…and a universal solution to treat any possible mutations in the given gene cannot be provided using NHEJ-based methodologies."

Reviewer 2 Report

In the review, Allen et al. described HDR-based genome engineering using CRISPR-Cas9 in CD34+ HSPCs. The gene editing nucleases, HDR mechanisms, donor delivery, and representative studies for correcting related genetic disorders were reviewed. Generally, the manuscript was well-written and informative. It could provide an overview of HDR-based gene therapy in blood and immune systems and would be of interest to a broad readership. The following comments are to further improve the manuscript.

Major comments:

1. Organization of the contents

Some organizations should be adjusted to improve overall coherence. For example, I suggest revising Table 1 to better reflect the information presented in the text. Currently, the donor platform is shown abruptly in the table without sufficient context, which could make the logic less clear for readers. The donor platform should be better to be referred in the latter sections.

2. Table 1 was mentioned in the introduction part for the disease-related gene corrections. I suggest revising it to provide more detailed information about diseases and pathogenic genes. Additionally, I recommend removing the ‘donor platform’ column and instead adding a column for pathogenic mutations. In the ‘Study’ column, it would be helpful to list the specific methods used, such as the type of cells, tissues, or animal models. This would provide readers with a more comprehensive understanding of the gene correction technologies employed in the studies mentioned in the table.

3. In Table 2, cell types and references should be added whenever possible.

Minor comments:

1. Brief introduction should be added for RAD51 when it shows first in line 176.

2. The full name of ‘DDR’ should be described when it showed first.

Author Response

Reviewer 2:

Major comments:

  1. Organization of the contents

Some organizations should be adjusted to improve overall coherence. For example, I suggest revising Table 1 to better reflect the information presented in the text. Currently, the donor platform is shown abruptly in the table without sufficient context, which could make the logic less clear for readers. The donor platform should be better to be referred in the latter sections.

We agree with the reviewer’s suggestion and have adapted the order accordingly. We moved Table 1 to the end of the manuscript and made Table 2 into Table 1. This way the old Table 1 is utilized more as a summary table and should be more clear for the reader.

  1. Table 1 was mentioned in the introduction part for the disease-related gene corrections. I suggest revising it to provide more detailed information about diseases and pathogenic genes. Additionally, I recommend removing the ‘donor platform’ column and instead adding a column for pathogenic mutations. In the ‘Study’ column, it would be helpful to list the specific methods used, such as the type of cells, tissues, or animal models. This would provide readers with a more comprehensive understanding of the gene correction technologies employed in the studies mentioned in the table.

We thank the reviewer for the comment. All the studies listed in Table 1 (now Table 2) were conducted in HSPCs, so we clarified it in the legend. In the “Study” column, “in vivo” referred to HSPCs edited ex vivo and transplanted to animal models. In addition, the scope of our review is gene editing by HDR, so, due to space limitations, we chose to omit the data regarding pathogenic mutations, mentioning only the affected gene.  

  1. In Table 2, cell types and references should be added whenever possible.

Since the focus of our manuscript is on studies performed in HSPCs, we omitted cell type-related information but mentioned HSPCs in the legend.

Minor comments:

  1. Brief introduction should be added for RAD51 when it shows first in line 176.

We agree with the reviewer’s suggestion and have added the following sentence: "RAD51 is an ATPase that creates a nucleoprotein filament on single-stranded DNA and locates and infiltrates homologous DNA sequences for the purpose of facilitating precise DNA repair.”

  1. The full name of ‘DDR’ should be described when it showed first.

We thank the reviewer for this catch and have corrected the mistake
